# Proportional Condylectomy Using a Titanium 3D-Printed Cutting Guide in Patients with Condylar Hyperplasia

**DOI:** 10.3390/cmtr18010007

**Published:** 2025-01-03

**Authors:** Wenko Smolka, Carl-Peter Cornelius, Katharina Theresa Obermeier, Sven Otto, Paris Liokatis

**Affiliations:** Department of Oral and Maxillofacial Surgery and Facial Plastic Surgery, University Hospital, LMU Munich, 80337 Munich, Germany; cornelius.peter@t-online.de (C.-P.C.); katharina.obermeier@med.uni-muenchen.de (K.T.O.); sven.otto@med.uni-muenchen.de (S.O.); paris.liokatis@med.uni-muenchen.de (P.L.)

**Keywords:** titanium 3D-printed cutting guide, proportional condylectomy, condylar hyperplasia

## Abstract

Background: The purpose of the study was to describe proportional condylectomy in patients with condylar hyperplasia using a titanium 3D-printed ultrathin wire mesh cutting guide placed below the planned bone resection. Methods: Eight patients with condylar hyperplasia underwent proportional condylectomy using an ultrathin titanium 3D-printed cutting guide placed below the planned bone resection. The placement of the guide was facilitated by the incorporation of anatomical landmarks. The accuracy of bone resections guided by such devices was evaluated on postoperative radiographs. The mean postoperative follow-up was 30 months. Results: Surgery could be performed in all patients in the same manner as virtually planned. The fitting accuracy of the cutting guides was judged as good. Postoperative radiographs revealed that the virtually planned shape of the newly formed condylar head after condylectomy could be achieved. Conclusions: In conclusion, the use of virtual computer-assisted planning and CAD/CAM-based cutting guides for proportional condylectomy in unilateral condylar hyperplasia of the mandible offers high accuracy and guarantees very predictable results.

## 1. Introduction

Condylar hyperplasia (CH) is caused by autonomous growth of the condylar head of the temporomandibular joint (TMJ) of unknown etiology [1,2,3,4]. Clinically, CH presents with progressive facial asymmetry of the lower face contour, distortion of the jawline, chin deviation towards the contralateral side, and malocclusion. If the onset of CH is during growth, maxillary compensation with canting of the maxilla and downward deviation of the occlusal plane to the ipsilateral side is common. If the onset of CH appears in adult patients, it usually results in either a unilateral posterior crossbite or an open unilateral posterior bite [5,6,7]. Often, CH presents in the second or third decade of life and mainly affects women [8].

A high condylectomy with or without disc replacement combined with simultaneous or second-stage orthognathic surgery has been initially postulated to be the treatment choice [9,10,11,12]. High condylectomy in patients with CH includes the removal of the upper 5 mm of the mandibular condylar head to remove its autonomous growing zone [10]. However, in recent years, a tendency to perform a proportional condylectomy can be observed [13]. Proportional condylectomy aims to remove the overgrown part of the affected condylar head, leading to the same shape and size of the condyle as the healthy non-affected side [8]. This surgical technique results in symmetry of the condylar processes on both sides and can prevent orthognathic surgery in CH patients with a unilateral posterior crossbite or an open unilateral posterior bite and an orthognathic maxilla. Furthermore, this surgical technique also simplifies orthognathic surgery in CH patients with canting of the maxilla and downward deviation of the occlusal plane to the ipsilateral side [13].

To achieve symmetric condylar processes in proportional condylectomy, precise preoperative planning is necessary. A reasonably high level of accuracy can be guaranteed by virtual computer-assisted planning and CAD/CAM-based cutting guides. Nowadays, this technique is already state-of-the-art in orthognathic surgery and bony reconstructive surgery [14,15].

The aim of the present study is to describe proportional condylectomy in patients with condylar hyperplasia using an ultrathin titanium 3D-printed cutting guide placed below the planned bone resection and to evaluate the outcome in the follow-up.

## 2. Materials and Methods

This study was approved by the institutional review board of the University Hospital of Munich (LMU Munich, Germany; No. 22-0445).

### 2.1. Demographics and Clinical Appearance

Between 2019 and 2023, 8 patients with condylar CH underwent proportional condylectomy using an ultrathin titanium 3D-printed cutting guide placed below the planned bone resection. The mean age was 31 years ranging from 19 years to 48 years. Two patients were male, and six were female. The reasons for treatment were malocclusion and functional problems in all cases. Two of all eight patients had an ipsilateral open bite without maxillary compensation.

The majority of patients (6 patients) presented clinically with maxillary compensation of the lateral movement of the mandible resulting in canting of the maxilla and downward deviation of the occlusal plane to the ipsilateral side. Five of these six patients had an additional ipsilateral open bite with a crossbite in two cases. Single photon emission computed tomography (SPECT) was performed in all patients, revealing an active form of CH in all cases, but no growth activity was found in the contralateral healthy condyle.

### 2.2. Titanium 3D-Printed Cutting Guide

For CAD/CAM production of the titanium 3D-printed cutting guide, each patient received a high-resolution computed tomography (CT) scan of the facial skeleton (isotropic resolution 0.625 mm), production of plaster models, and occlusal scans of the corresponding dental arch. Resulting DICOM data from CT scans and stereolithography (STL) data from occlusal scans were sent to an industrial partner for subsequent virtual surgical planning using ProPlan CMF software ProPlan Version 3.0.1.5. (Materialise^®^, Leuven, Belgium). Virtual surgical planning was performed in an interactive online meeting with the clinical engineers and the surgeon.

In the first planning step, the healthy condylar head of the contralateral side was mirrored to the affected side to identify the dimension of a regular-shaped condylar head and to determine the amount of bone that was needed to be resected for adequate proportional condylectomy (Figure 1). This first step was only a rough estimation as both condylar heads are not absolutely symmetrical. In the next planning step, cutting planes were virtually created to simulate the resection. After that, the resulting condylar head was virtually repositioned into the joint fossa, and the angle of the cutting plane was corrected if necessary. In cases without simultaneous orthognathic surgery, the occlusion after virtual repositioning of the condylar head was also used as a reference.

A titanium 3D-printed cutting guide was preferred because of its smaller dimensions compared with a plastic one and, therefore, easier intraoperative placement, allowing accurate placement of the guide without extensive stripping of soft tissues. Its mesh design also allows visual control of the correct adaptation. The extent of the cutting guide was planned according to anatomical landmarks, such as the remaining lateral pole of the condylar head, posterior border of the condylar process, and sigmoid notch, in order to guarantee accurate placement of the guide during surgery (Figure 2). A screw hole was designed for fixation of the cutting guide with a screw of 2 mm in diameter to avoid rotation of the guide during condylectomy. After approval of the planned design, the cutting guide went into computer-aided manufacturing using resin-based or selective laser melting.

In cases where bone resection was necessary in a second plane—for example, resection of bone on the lateral pole of the condylar head—a second cutting guide with screw holes in the identical position as the first guide was produced for accurate placement of the second guide (Figure 3).

### 2.3. Surgical Procedures

Surgical treatment was performed under general anesthesia in all patients. A preauricular approach was used in all patients to expose the TMJ area and to place the cutting guide. Bone resection was performed with a piezo-electric surgery device (Mectron, Cologne, Germany). In patients with an open bite only (n = 2) and no canting of the maxilla, intraoperative mandibulo-maxillary fixation (MMF) using titanium arch bars (Medartis AG, Basel, Switzerland) was performed in order to control the occlusion and the proper position of the newly created condylar head in the fossa intraoperatively (Figure 4).

Functional postoperative treatment with elastics was only administered if there were minor occlusal disturbances due to postoperative swelling. No orthodontic treatment was necessary to achieve a sufficient occlusion.

In patients with CH and maxillary compensation with canting of the maxilla and downward deviation of the occlusal plane to the ipsilateral side (n = 6), simultaneous orthognatic surgery was performed. Virtual planning of orthognathic surgery for manufacturing of CAD/CAM cutting guides, surgical splints, as well as patient-specific implants (PSIs) took place in the same planning session as for planning of the proportional condylectomy (Figure 5). In this patient group, all patients (n = 5) received simultaneous orthodontic treatment. Immediately after the operation, the fitting accuracy of the cutting guide, as well as the intraoperative handling, was categorized by the main surgeon as good, moderate, or poor.

### 2.4. Postoperative Clinical and Radiological Evaluations

Postoperative clinical and radiological data were evaluated in a retrospective manner. For the postoperative radiographic outcome assessment, all patients underwent panoramic X-rays as well as CT scans. Radiological follow-up after 12 months was performed using panoramic X-rays in 2 patients and a CT scan in 1 patient. These 3 patients were orthognathic cases, and radiographs were taken due to planning of plate removal. In the remaining 5 patients, no radiological follow-up was performed due to excellent clinical outcomes in terms of mandibular movements and occlusion.

Postoperative clinical follow-up appointments took place up to a mean of 30 months (range of 6 to 48 months). All kinds of complications occurring within this time frame, such as recurrence, wound infection, wound dehiscence, malocclusion, facial nerve weakness, and palsy, were recorded.

## 3. Results

Surgery could be performed on all patients in the same manner as virtually planned. The fitting accuracy of the cutting guides for proportional condylectomy was judged as good by the performing surgeons in all cases. Also, intraoperative handling of the cutting guide was rated to be good in all eight patients. In the two patients with open bite only and no canting of the maxilla, proper habitual occlusion could be achieved without any orthodontic treatment. These patients were treated postoperatively using titanium arch bars and elastics for 2 weeks. One further patient who received orthognatic surgery was treated postoperatively with elastics for 2 weeks. In six patients with CH and maxillary compensation with canting of the maxilla, repositioning of the newly formed condylar head after proportional condylectomy into the fossa during orthognathic surgery was possible without any interference. The fitting accuracy of the upper and lower teeth into the virtually planned surgical splints after bimaxillary osteotomies and internal fixation was sufficient and without any interference.

Postoperative radiographic assessment using panoramic X-rays as well as CT scans revealed that a virtually planned shape of the newly formed condylar head after proportional condylectomy could be achieved in all patients. No signs of recurrence could be found in the radiological follow-up of three patients after 12 months, with panoramic X-rays performed in two patients and CT scans in one patient. In the remaining five patients, no radiological follow-up was performed due to excellent clinical outcomes in terms of occlusion, mandibular movements, and facial symmetry.

Postoperatively, no wound infections or wound dehiscences were reported. Malocclusion was not present in any patient. Neither facial nerve weakness nor facial nerve palsy were observed. After a follow-up of 30 months, clinically, no recurrence occurred.

## 4. Discussion

Unilateral CH often requires surgical treatment, especially in patients with an active form of CH [9,10,11,12]. In the current article, we describe digital planned proportional condylectomies in eight patients using titanium 3D-printed cutting guides placed under the osteotomy line in eight patients with active CH.

Two patients without maxillary canting were treated with condylectomy alone, and sufficient occlusion, mandibular movements, and facial symmetry could be achieved. Similarly, in the six patients receiving proportional condylotomies combined with orthodontic treatment, equally good clinical and radiological findings were observed postoperatively. The present study demonstrates that virtual computer-assisted planning and CAD/CAM-based cutting guides applied on the healthy bone under the osteotomy guarantee a high accuracy for proportional condylectomy, leading to predictable results, no recurrence, and reducing the need for further surgeries.

Treatment with condylectomy without orthognathic surgery of patients with CH and no maxillary canting has been reported before. Figueroa et al. presented three cases of patients treated with cutting guides placed under the osteotomy line [16]. However, they used guides from stereolithographic resin that were often quite bulky and covered the whole bone surface, which could lead to difficult or not precise placement. In contrast to this, the titanium cutting guides shown in the present study are quite thin, and because of the mesh design, they allow for a good evaluation of the fit on the bone surface. Furthermore, Figueroa et al. showed that this treatment option could result in sufficient facial symmetry but did not report on the activity status of CH. Finally, in our study, the cutting guides were used in combination with digitally planned orthognathic surgery, demonstrating the efficiency of digitally planning the osteotomy together with the orthognathic surgery.

Sembronio et al. reported on computer-guided proportional condylectomy using custom-fitted 3D-printed surgical cutting guides from resin [17,18]. They placed the guides on the portion of bone that needed to be resected. In the author’s experience, however, this technique only works for extensive resections, as the space for placing the screws is otherwise limited. It will also be impractical if the major part of the bone needed to be removed is located on the medial pole of the condylar head.

Regarding the timing of the surgical intervention in patients with CH, before or after the active phase of CH, some surgeons prefer to wait until the stabilization of the disease. In our case series with confirmed active CH, we show that early surgery supported with 3D-cutting guides can result in predictable results and decrease the severity of facial deformities, as well as the number of operations needed [19,20].

Finally, an intraoral approach for computer-guided proportional condylectomy using a CAD/CAM surgical cutting guide has also been described [21,22]. However, this approach only offers a limited overview of the surgical field. It, therefore, seems to be only appropriate for an extensive lower resection of the condylar head. In addition, it also seems to be insufficient if the main part of the hyperplastic bone is located on the medial condylar pole or posterior to the condylar head. In contrast, in the present study, the use of titanium mesh guides and a preauricular approach offers an excellent overview of the surgical field, easy handling of the cutting guide, and high accuracy.

Overall, this study still has some limitations. The study design is retrospective, and a postoperative radiological assessment using 3D radiographs is missing as most of the patients did not receive postoperative CT scans. Therefore, the accuracy of the presented surgical method could only be assessed by clinical examination. In addition, because of the low incidence of the disease, the study only included a small number of patients. A prospective study design with consequent postoperative 3D radiological assessment might help to evaluate the accuracy better. In the future, multicenter studies might also help to increase the number of patients included in such a study.

## 5. Conclusions

In conclusion, the use of virtual computer-assisted planning and CAD/CAM-based cutting guides for proportional condylectomy in unilateral CH offers high accuracy and guarantees very predictable results.

## Figures and Tables

**Figure 1 cmtr-18-00007-f001:**
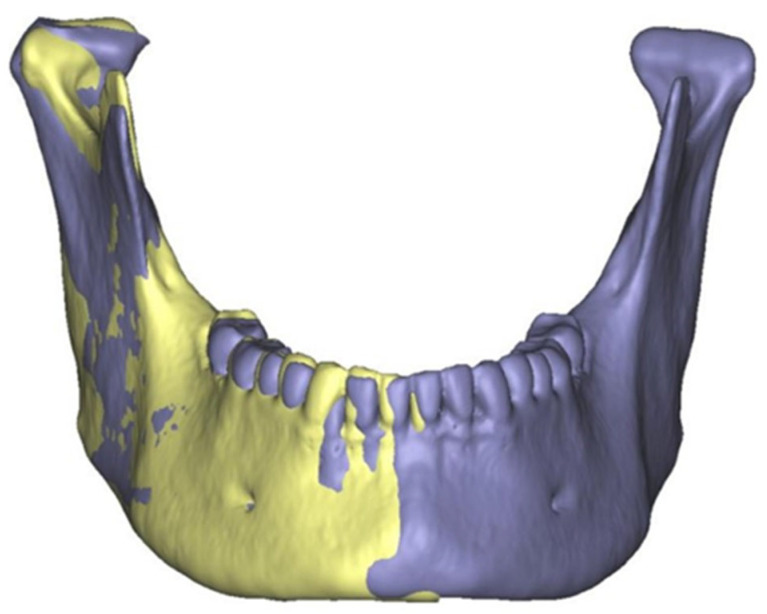
Computer-assisted virtual planning of a proportional condylectomy of the right mandibular condyle. The original entire mandible with the distorted right condylar head is shown in purple. The healthy left hemi-mandible in yellow color has been horizontally mirrored and superimposed on the affected side to indicate the amount of resection needed to equal healthy conditions.

**Figure 2 cmtr-18-00007-f002:**
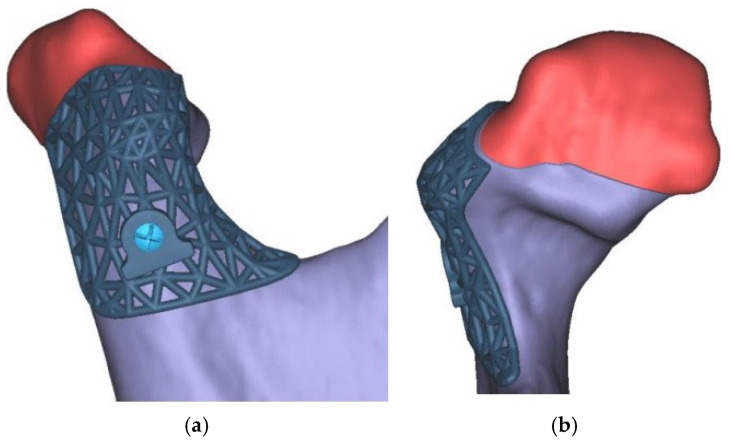
(**a**) Virtual planning of a standalone cutting guide in lateral view (**b**) and frontal view. The portion of bone that is aimed to be resected is marked in red.

**Figure 3 cmtr-18-00007-f003:**
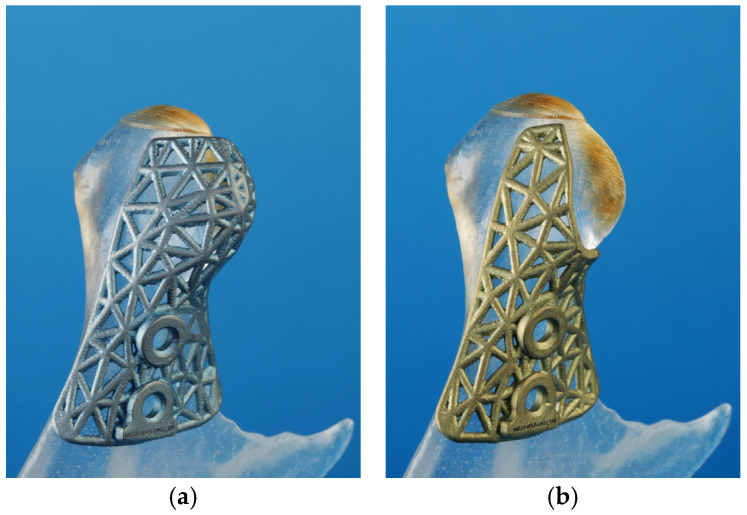
Paired cutting guides for a combined stepwise proportional condylectomy procedure: (**a**) initial cutting guide to resect the cranial portion of the condylar head; (**b**) second cutting guide for resection of the lateral portion of the condylar head. Both guides have a plate hole for screw fixation in a coinciding bony drill hole.

**Figure 4 cmtr-18-00007-f004:**
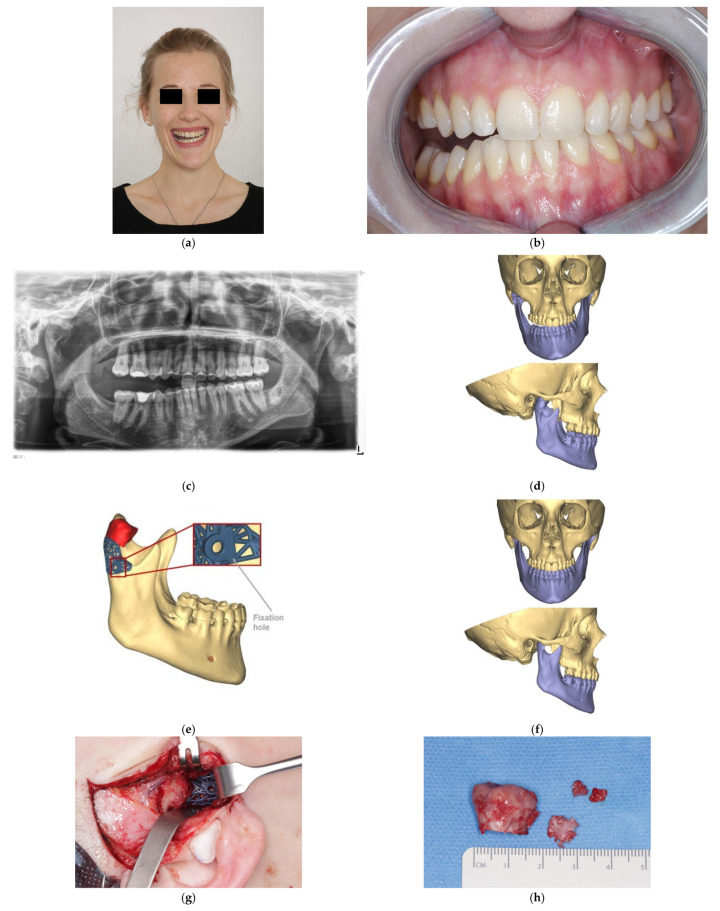
Patient with right CH and ipsilateral open bite. (**a**) Preoperative frontal view, (**b**) occlusion, and (**c**) panoramic X-ray. (**d**) Virtual planning preoperative frontal and lateral view. (**e**) Virtual planned cutting guide. (**f**) Virtual planned postoperative frontal and lateral view. (**g**) Intraoperative view with cutting guide. (**h**) Resected bone. (**i**) Intraoperative MMF. (**j**) Coronal plane of postoperative CT scan. (**k**) Postoperative occlusion. (**l**) Postoperative frontal view.

**Figure 5 cmtr-18-00007-f005:**
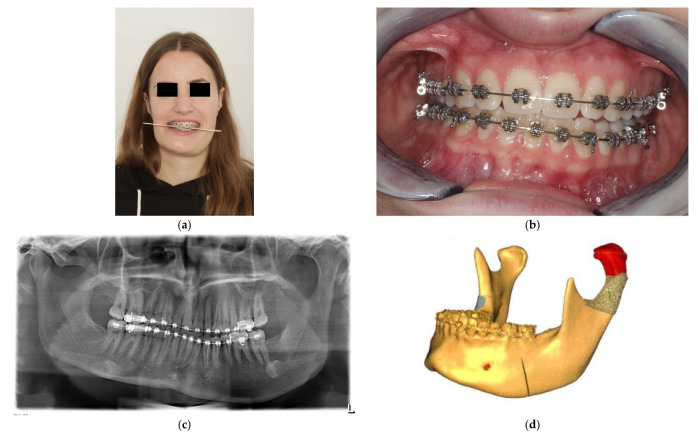
Patient with left CH and maxillary compensation with canting of the maxilla and downward deviation of the occlusal plane to the ipsilateral side. (**a**) Preoperative frontal view, (**b**) occlusion, and (**c**) panoramic X-ray. (**d**) Virtual planned cutting guide. (**e**) Virtual planned orthognatic surgery—Le Fort I Osteotomy. (**f**) Virtual planned postoperative frontal view. (**g**) Intraoperative view with cutting guide for condylectomie. (**h**). Resected bone. (**i**) Postoperative panoramic X-ray. (**j**) Postoperative occlusion. (**k**) Postoperative frontal view.

## Data Availability

The data presented in this study are available on request from the corresponding author.

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
