# Peer review of "Proportional Condylectomy Using a Titanium 3D-Printed Cutting Guide in Patients with Condylar Hyperplasia"

_1943-3883, 2025, doi:10.3390/cmtr18010007_

Round 1
Reviewer 1 Report
Comments and Suggestions for Authors
This is a case series reporting the results of a modification/refinement of existing technology to treat condylar hyperplasia. whilst it is a case series and has the inherent weaknesses associated with any case series report, it does describe an improvement in technique for managing these cases and it does therefore contribute to the field of knowledge in this area. I believe it should be published in the Journal.
Author Response
Comment: This is a case series reporting the results of a modification/refinement of existing technology to treat condylar hyperplasia. whilst it is a case series and has the inherent weaknesses associated with any case series report, it does describe an improvement in technique for managing these cases and it does therefore contribute to the field of knowledge in this area. I believe it should be published in the Journal.
Answer: Thank you very much for recommending our paper for publication.
Reviewer 2 Report
Comments and Suggestions for Authors
Dear authors
I would like to congregate you for your work
however, I would like to concentre more on the value of your surgical guide on the results of your surgical work
and the value of your article could be more efficient if you add a control group to compare the accuracy of your surgical guide and the impact of the guide on operation time and the proportional resection of the condyle
Author Response
Comments 1: I would like to concentre more on the value of your surgical guide on the results of your surgical work.
Response 1: Thank you very much for your comment. However, we think that there is a strong focus on the value of our surgical guide in the results. We descibe the fitting accuracy, intraoperative handling handling as well as the postoperative outcome that could be achieved with these guides.
Comment 2: The value of your article could be more efficient if you add a control group to compare the accuracy of your surgical guide and the impact of the guide on operation time and the proportional resection of the condyle.
Response: This is a retrospective study. There is no such control group available for us.
Reviewer 3 Report
Comments and Suggestions for Authors
Please clarify what studies were done to ascertain the completion of growth before attempting to address the condyles.
In case growth was still ongoing since some of the patients were 19 why was the condyle adressed and what postoperative review was done to check for a recurrence of the problem?
References are not cited as per Vancouver style
Author Response
Comments:Please clarify what studies were done to ascertain the completion of growth before attempting to address the condyles. In case growth was still ongoing since some of the patients were 19 why was the condyle adressed and what postoperative review was done to check for a recurrence of the problem?
Response: We added to line 69/70 the following explanation: Single photon emission computed tomography (SPECT) has been performed in all patients revealing an active form of CH in all cases, whereas no growth activity was found in the contralateral healthy condyle. We described already the "check for a recurrence" in the section "Postoperative Clinical and Radiological Evaluation".
